# Adipose Tissue Inflammation and Pulmonary Dysfunction in Obesity

**DOI:** 10.3390/ijms23137349

**Published:** 2022-07-01

**Authors:** Giuseppe Palma, Gian Pio Sorice, Valentina Annamaria Genchi, Fiorella Giordano, Cristina Caccioppoli, Rossella D’Oria, Nicola Marrano, Giuseppina Biondi, Francesco Giorgino, Sebastio Perrini

**Affiliations:** Section of Internal Medicine, Endocrinology, Andrology and Metabolic Diseases, Department of Emergency and Organ Transplantation, University of Bari Aldo Moro, 70124 Bari, Italy; giuseppe.palma@uniba.it (G.P.); sorice.gianpio@gmail.com (G.P.S.); valentina.genchi@uniba.it (V.A.G.); fiorella.giordano01@gmail.com (F.G.); cristina.caccioppoli@uniba.it (C.C.); rossella.doria@uniba.it (R.D.); nicola.marrano@uniba.it (N.M.); giuseppina.biondi@uniba.it (G.B.); francesco.giorgino@uniba.it (F.G.)

**Keywords:** obesity, respiratory diseases, lung, adipose tissue, crosstalk, adipokines, inflammasome

## Abstract

Obesity is a chronic disease caused by an excess of adipose tissue that may impair health by altering the functionality of various organs, including the lungs. Excessive deposition of fat in the abdominal area can lead to abnormal positioning of the diaphragm and consequent reduction in lung volume, leading to a heightened demand for ventilation and increased exposure to respiratory diseases, such as chronic obstructive pulmonary disease, asthma, and obstructive sleep apnoea. In addition to mechanical ventilatory constraints, excess fat and ectopic deposition in visceral depots can lead to adipose tissue dysfunction, which promotes metabolic disorders. An altered adipokine-secretion profile from dysfunctional adipose tissue in morbid obesity fosters systemic, low-grade inflammation, impairing pulmonary immune response and promoting airway hyperresponsiveness. A potential target of these adipokines could be the NLRP3 inflammasome, a critical component of the innate immune system, the harmful pro-inflammatory effect of which affects both adipose and lung tissue in obesity. In this review, we will investigate the crosstalk between adipose tissue and the lung in obesity, highlighting the main inflammatory mediators and novel therapeutic targets in preventing pulmonary dysfunction.

## 1. Introduction

It is well established that obesity has reached epidemic proportions, especially in Western industrialised nations, and that it has become a leading global cause of disease and death. Obesity has serious implications for the health and functioning of the respiratory system, where it results in increased prevalence, morbidity, and clinical presentation of several diseases [1]. As the global morbidity rate of chronic respiratory diseases is significantly increasing, the influence of obesity on respiratory diseases and vice versa requires further intense study [2].

There are obvious mechanical effects of obesity on lung function that explain the associations between obesity and lung diseases, especially regarding obesity hypoventilation syndrome (OHS) and obstructive sleep apnoea syndrome (OSAS) [1,2,3,4,5], in addition to the mechanical effects of obesity that influence asthma and pulmonary hypertension [1,2,6,7].

Furthermore, obesity affects outcomes in acute respiratory distress syndrome (ARDS) and chronic obstructive pulmonary disease (COPD) [8,9]. The rise in the prevalence of obesity affects the epidemiology of pulmonary diseases. Several studies have demonstrated that susceptibility to respiratory infections and hospitalisation rates are higher in obese patients than healthy-weight subjects [1,2,10,11,12]. Chronic inflammatory diseases of the respiratory system develop based on the understanding of obesity as a state of chronic inflammation [1,11,13]. Adipose tissue, formerly considered a passive repository organ, is now increasingly recognised as a large endocrine organ with an active role in energy homeostasis. The endocrine functions of adipose tissue depend on its ability to secrete hormones, factors, and protein signals called ‘adipokines’ responsible for its metabolic and pro-inflammatory effects. In obesity, functional dysfunction of adipose tissue leads to increased release of pro-inflammatory adipokines, affecting metabolic function in addition to lung health. Therefore, investigating the crosstalk between adipose tissue and the lung and understanding the interactions between obesity and lung disease is of much interest [2,14,15,16].

Although the association between adipose tissue dysfunction and lung disease has previously been established by several studies [17,18,19,20], the precise crosstalk mechanism remains unclear. Emerging evidence has suggested that the NLR family pyrin domain-containing 3 (NLRP3) inflammasome, a multimeric protein involved in inflammation, could be the main inducer of a cytokine cascade with multiple detrimental effects on several organs, including adipose tissue and the lungs [21,22,23,24].

This review discusses how obesity affects the normal physiology and function of the lungs, in addition to how it leads to the pathophysiology of pulmonary diseases, such as COPD, asthma, and OSAS. Moreover, we define the molecular mechanism underlying the crosstalk between adipose tissue and the lung in obesity to highlight potential signalling endpoints that could be used as therapeutic targets for the treatment of lung diseases induced by obesity.

## 2. The Mechanical Effects and the Impact of Obesity on Lung Function

The impact of obesity on respiratory function and the increased burden of respiratory diseases among obese people are significant [1,2,25]. The impact begins with the critical distribution of fat, which represents a mechanical obstruction of lung function. Android obesity (i.e., fat distributed in the thorax, abdomen, and visceral organs) has a more direct effect on pulmonary function than gynoid obesity (i.e., fat distributed in the subcutaneous tissue of the hips, thighs, arms, and legs) [1]. The major impact of android obesity on pulmonary function depends on increased abdominal volume and the presence of intrathoracic fat that fosters the displacement of the diaphragm, reducing lung volumes, specifically functional residual capacity (FRC) and expiratory reserve volume (ERV) [1,25]. FRC reduction is directly proportional to the severity of obesity, with overweight, mildly obese, and severely obese subjects demonstrating reductions of 10%, 22%, and 33%, respectively [1]. In obese subjects, the mechanical encumbrance caused by excess adipose tissue reduces the calibre of the airways, resulting in limited expiratory flow and low lung volumes [25]. One of the mechanical complications of obesity is greater respiratory system stress due to constriction of the airways [1]. Recent studies have revealed a strong association between obesity and pulmonary hypertension, a condition characterised by an increase in mean pulmonary arterial pressure [7]. Alterations in respiratory system mechanics caused by obesity include expiratory flow limitation, atelectasis, and V/Q mismatch (occurring when either the ventilation airflow or perfusion blood flow is impaired, limiting the primary lung function of delivering oxygen to the blood) with hypoxaemia, all of which have important implications in the context of critical illness [10]. Tidal volume is also slightly lower in obese individuals; however, there is no significant effect on residual volume (RV) or total lung capacity (TLC) [1]. Fat accumulations in the thorax and abdomen have been found to result in an increase in lung elastance of up to 35% in obese individuals, which is exponentially related to body mass index (BMI) [9]. This reduced compliance is a consequence of significant alterations in the mechanical properties of the whole respiratory system in the presence of fat deposits in the mediastinum and abdominal cavities; it contributes to respiratory difficulties, like wheezing, shortness of breath (dyspnoea), and breathlessness while sleeping (orthopnoea) [1]. However, the mechanical properties of the chest wall in obese subjects indicate chest mass loading rather than chest mass stiffening by elastic loading, which complicates the simplistic association between a constrictive effect and fat volume and may indicate that fat volume increases compliance [9].

Several studies have shown that obese individuals are at greater risk of developing ARDS, a life-threatening lung injury that allows fluid to leak into the lungs [26]. This observation was highlighted during the recent coronavirus disease 2019 (COVID-19) pandemic, a viral disease that has a significant impact on lung health [27]. Although obese subjects are the least likely to die from this condition for reasons that are not fully understood [1,2], managing ARDS in obese patients presents significant challenges, especially in the context of intubation and respiratory support [9,10].

## 3. Adipose Tissue Dysfunction and Pulmonary Diseases

In addition to the mechanical effects described above, obesity is responsible for chronic low-grade inflammation caused by adipocyte-derived signal mediators that have a negative impact on pulmonary function [11]. The function of adipose tissue extends beyond energy storage; it is responsible for secreting several soluble factors, known as adipokines, that act in an autocrine and paracrine manner to regulate the function of the adipose tissue itself, and in an endocrine manner on different target organs, such as the heart and lungs [28]. Moreover, adipokines can regulate secretion of cytokines in other tissue with pro- or anti-inflammatory effects, depending on the biological context [29,30,31,32].

In humans, adipose tissue can be found in the subcutaneous layer (‘subcutaneous adipose tissue’, SAT) or around internal organs (‘visceral adipose tissue’, VAT). Recently, visceral depots of fat have also been found in the lungs, where lipids are stored in lipid-laden adipocyte-like cells known as ‘lipofibroblasts’. These cells have a key role in regulating alveolar lipid homeostasis and pulmonary surfactant production, facilitating oxygen absorption [33].

Under physiological conditions, lipids are deposited preferentially in SAT, while VAT and lung lipofibroblasts normally contain limited quantities of lipids. However, in obesity, adipose tissue may become dysfunctional and not expand adequately to store the excess energy [34]. Moreover, SAT has a limited capacity to expand, which is genetically predetermined, and it can become saturated in morbid obesity. In our previous work, we showed that nutritional overload in obesity may enhance the expandability of VAT, specifically over SAT, which may occur via downregulation of expression of the histone deacetylases sirtuin 1 (SIRT1) and sirtuin 2 (SIRT2) in the resident adipose staminal cells (ASCs) [35]. Under these conditions, lipids are mainly deposited in VAT [35,36,37,38], a process referred to as ‘ectopic fat accumulation’, which can promote a low-grade systemic inflammation, the harmful effects of which can alter cardiometabolic function and lung health [39,40]. Ectopic deposition of fat can also occur in the lung in obesity, obstructing airways and worsening pulmonary damage and prognosis in patients with COVID-19 [41,42].

In healthy non-obese subjects, caloric excess determines SAT expansion mainly via hyperplasia. Conversely, pathological expansion of VAT in morbid obesity largely results from adipocyte hypertrophy due to the dimensional increase of intracellular lipid droplets, promoting systemic inflammation [34]. Larger lipid droplets in hypertrophic adipocytes can disrupt the function of cortical actin filaments involved in the insulin-stimulated transfer of glucose transporter type 4 (GLUT4) to the plasma membrane, impairing insulin-dependent glucose uptake in the VAT of morbidly obese subjects [43]. Moreover, adipocyte hypertrophy can promote activation of intracellular pro-inflammatory signalling pathways, altering the adipocyte secretory profile [44,45]. There are clear differences in the secretome release between SAT and VAT, and we have previously demonstrated that this biological specificity is already established in ASCs from these depots [46]. In this regard, several studies have shown that VAT secretes more pro-inflammatory mediators than SAT in morbid obesity, leading to a systemic low-grade inflammation that is responsible for an increased risk of developing insulin resistance, hepatic steatosis, cardiometabolic diseases, and pulmonary dysfunction [47,48,49]. Under these conditions, adipose tissue becomes dysfunctional because it becomes less insulin responsive and secretes more pro-inflammatory adipokines, promoting macrophage infiltration and further aggravating insulin resistance and lung diseases [50]. Particularly, expression of leptin and IL-6 appears to be increased in VAT of obese subjects with asthma, thus suggesting that these mediators could promote airway inflammation [17].

As in VAT adipocytes, ectopic lipid accumulation can also occur in lung lipofibroblasts in obesity, altering their function. When alveolar lipid homeostasis is disrupted in morbid obesity, dysfunctional lipofibroblasts could transdifferentiate into myofibroblasts, which produce excess extracellular matrix that can lead to pulmonary fibrosis [33].

Another harmful consequence of pathological VAT expansion is adipose tissue hypoxia. Under physiological conditions, increased angiogenesis occurs as adipose tissue expands, ensuring adequate oxygen and nutrient supply; however, the VAT of morbidly obese subjects has a reduced capacity to create new vessels through angiogenesis as compared to SAT [51]. As adipocyte hypertrophy progresses in VAT, local tissue hypoxia develops, resulting in the activation of hypoxia-inducible transcription factors and increasing the production of reactive oxygen species (ROS) [52,53]. The imbalance between ROS generation and antioxidant defence systems represents the primary cause of endothelial dysfunction in several tissue types, leading to pulmonary and atherosclerotic diseases [54,55]. Many of the maladaptive processes and extrapulmonary comorbidities of lung diseases have been linked to tissue hypoxia, which is responsible for changes in cellular metabolism, inducing local insulin resistance in muscle and adipose tissue and perpetuating the molecular basis of peripherally reduced insulin action [8].

It is clear that obese subjects with a greater genetic predisposition to expand SAT are less exposed to the systemic low-grade inflammation induced by obesity. Of note, it has been shown that obese subjects with a lower VAT/SAT ratio have a reduced risk of cardiometabolic disease and better lung function as compared with subjects with a higher VAT/SAT ratio [19,20,56]. This type of obesity has recently been classified in the literature as ‘metabolically healthy obesity’ (MHO) and is defined by the absence of metabolic disorders and cardiovascular disease [57]. MHO subjects have a lower VAT/SAT ratio, gynoid rather than android fat distribution, reduced levels of pro-inflammatory adipokines, and a reduced risk of cardiometabolic dysfunction [57]. Metabolic health is also associated with lung function, and this association appears to be stronger than the association between obesity and lung function. In contrast, recent studies have demonstrated that ‘metabolically unhealthy obesity’ (MUHO) subjects with a higher VAT/SAT ratio were more prone to decreased lung function as compared with their metabolically healthy counterparts, regardless of their obesity status [58]. Therefore, identification of both adiposity subgroups is necessary and useful in predicting the development of pulmonary diseases.

These findings indicate that metabolic alterations and pulmonary diseases could have a common pathogenic mechanism that involves low-grade systemic inflammation triggered by adipose tissue dysfunction.

## 4. Obesity-Associated Pulmonary Diseases

### 4.1. Pulmonary Infections and Obesity

Compared with nonobese controls, obesity has been associated with a fivefold higher risk of hospitalisation and a threefold higher risk of death due to pulmonary infectious diseases [9], as obesity increases susceptibility to infectious pathogens, particularly viral infections [11]. This susceptibility is due to secretory changes within the expanded VAT mass that result in chronic low-grade inflammation [11]. Due to chronic low-grade inflammatory preconditioning, adipose tissue is also a source of many pro-inflammatory mediators and adipokines that may enhance the characteristic cytokine storm, lastly seen in COVID-19 infection [12]. Recent evidence has suggested that dysfunctional lipofibroblasts in obesity may play a key role in the pulmonary fibrosis induced by viral infections [59]. Lipofibroblast activity is regulated by lung epithelial cells that express high levels of angiotensin-converting enzyme 2 (ACE-2), a surface protein that can be used as an entry receptor by several respiratory viruses [60]. Injury to lung epithelial cells caused by viral infection can induce transdifferentiation of lipofibroblasts to myofibroblasts, leading to lung fibrosis. As previously discussed, obesity has a negative impact on lipofibroblast function, rendering the cells more prone to profibrotic transition. Based on these findings, it has been hypothesised that obesity could be a negative prognostic factor in patients with COVID-19, as ectopic fat deposition in the lungs of obese subjects could lead to lipofibroblast dysfunction, promoting lung fibrosis [59].

### 4.2. Obstructive Sleep Apnoea Syndrome and Obesity

OSAS is a common health condition that has a major impact on public health globally [15,52]. OSAS and obesity have been shown to have negative effects on a variety of organs and systems [15,52,61]. OSAS is associated with obesity in more than 60% of cases and is linked to the development of several comorbidities, including hypertension, arrhythmia, stroke, coronary heart disease, and metabolic dysfunction [52]. Recently, it was recognised that the gut microbiota has a role in the development of OSAS and obesity and might thus be a factor in the coexistence of OSAS and obesity [15]. The pathogenesis of cardiometabolic problems in OSAS is unknown; however, the pattern of intermittent hypoxia (IH) seen in OSAS, which involves repeated, brief cycles of desaturation and reoxygenation, is likely to play a role [52]. IH appears to mediate some of its negative effects through adipose tissue inflammation and malfunction [52]. Macrophages in fat (specifically M1 rather than M2 phenotype) are likely the target cells for the effects of persistent IH, resulting in elevated inflammatory biomarkers, which suggests that OSAS and obesity may have synergistic effects [15,52]. In this regard, we recently demonstrated that subjects with OSAS and obesity had increased exposure to glycaemic control impairment and adipose tissue inflammation, underlying the relationship between metabolic dysfunction and the development of pulmonary diseases [4,5]. These findings were further supported by evidence that administration of continuous positive airway pressure therapy (cPAP) in obese subjects with OSAS reduced inflammatory markers in both adipose tissue and sera, suggesting that therapeutic correction of IH combined with weight loss interventions has the potential to reduce cardiometabolic risk [5]. Given the pathophysiological causes of IH and sleep fragmentation in OSAS, sympathetic activation, oxidative stress, inflammation, and metabolic dysregulation are all potential pathways of OSAS and obesity [15]. Furthermore, a growing body of research suggests that hypoxia plays a significant role in cancer as a low-grade inflammatory state that represents a common origin for neoplastic diseases [52].

### 4.3. Chronic Obstructive Pulmonary Disorder and Obesity

As in the case of OSAS, obesity has been linked to COPD, a type of chronic lung disease characterised by obstructed flow through respiratory airways. It is now known that metabolic syndrome is a comorbidity in up to 50% of COPD patients, and insulin resistance (the most relevant clinical feature of metabolic syndrome) is a trademark of excess adipose tissue [62]. When compared with the general population, COPD patients had double the rate of metabolic syndrome, with prevalence ranging from 21% to 62%, and over half of COPD patients showed signs of one or more metabolic syndrome condition [8]. Importantly, COPD comorbidities, such as obesity, have a major impact on quality of life of patients and significantly affect health-care utilisation [63]. Excess abdominal fat, BMI ≥ 30 kg/m^2^, increased blood pressure, a proatherogenic blood lipid profile, and impaired fasting blood glucose with or without insulin resistance are all clinical signs of metabolic syndrome [8]. It is still unclear whether metabolic syndrome is a separate, coexisting illness or a direct result of worsening lung pathology in COPD patients.

### 4.4. Asthma and Obesity

Asthma is a long-term inflammation of the respiratory airways characterised by airway hyperresponsiveness (AHR), defined as the predisposition of the airways to narrow excessively in response to stimuli [64]. It has been demonstrated that obesity is often a comorbidity of asthma, as bariatric surgery performed in obese asthmatic patients was shown to alleviate symptoms and reduce airway inflammation [17]. Moreover, patients with both obesity and asthma have an increased risk of aggravation and a decreased response to treatment with inhaled and systemic corticosteroid therapies as compared with lean asthmatic patients [6]. Obese asthmatic patients have been classified into several phenotypes. Asthmatic patients with earlier-onset disease, where asthma has been present since adolescence and worsened with the onset of obesity, have the most severe symptoms, with strong airway inflammation mediated mainly by T-helper 2 (Th2) immune cells. In asthmatic patients with later-onset disease, where asthma was de novo and arose as a consequence of obesity, inflammation is milder in the lung, but is exacerbated in adipose tissue. Finally, there is perhaps a separate group in which obese subjects have an increased susceptibility to environmental pollutants; however, it is not clear whether this contributes to a distinct phenotype or complicates other phenotypes [65]. Therefore, although obesity may not cause asthma in some cases, there is clear evidence that obesity can worsen asthma symptoms and reduce response to asthma treatment.

### 4.5. Obesity Hypoventilation Syndrome

OHS is the result of severe obesity combined with hypoventilation [11]. The prevalence of OHS has been estimated at 0.4% of the adult population, and the contribution of OHS to morbidity and mortality is likely to increase as the prevalence of obesity rises [66,67]. It is therefore critical that the condition is clearly defined so that patients can be rapidly identified [66].

When symptoms lead to pulmonary or sleep consultation under stable conditions, OHS is usually diagnosed during an episode of acute-on-chronic hypercapnic respiratory failure. The definitive test for alveolar hypoventilation is room-air arterial blood gas, and the diagnosis is confirmed after arterial blood gases and a sleep study [67]. After ruling out other disorders that may cause alveolar hypoventilation, OHS is defined as a combination of obesity (BMI ≥ 30 kg/m^2^), daytime hypercapnia (arterial carbon dioxide tension > 45 mmHg), and disordered breathing during sleep [67].

Excess adipose tissue in the abdomen and surrounding the chest wall reduces lung volume, namely FRC, with a significant decrease in the ERV [67]. Obese patients have a chest wall and total respiratory compliance of 92–80%, while obese patients who also have OHS have a compliance of 44–37%. Furthermore, obese people, particularly OHS patients, have increased airway, chest wall, and respiratory system resistance, which may be due to a decrease in lung volume [11]. Fat deposits have direct mechanical effects on respiratory function by impeding diaphragm motion, reducing lung compliance, and increasing lower airway resistance. Gas that is trapped due to premature airway closure generates intrinsic positive end-expiratory pressure and favours V/Q mismatch, leading to the development of atelectasis in the lower lobes of the lungs. Several coexisting mechanisms, such as obesity-related changes in the respiratory system, changes in respiratory drive, and breathing abnormalities during sleep, explain the presence of daytime hypercapnia [67].

Patients with OHS have worse respiratory mechanics than those who are morbidly obese but do not have OHS, in addition to having respiratory muscle weakness. The work required for breathing is increased and must be compensated by increased drive from the respiratory centres to the respiratory muscles [67]. Severe obesity may also increase the metabolic demand of breathing. Respiratory muscle demand accounts for roughly half of the 60% increase in resting oxygen consumption among obese patients (mean BMI 53 kg/m^2^) as compared with control subjects [1]. This increased ventilatory load causes a compensatory increase in neural respiratory drive, a mechanism that is disrupted in OHS, resulting in decreased neural drive, hypercapnia, and hypoxaemia [1].

## 5. Mechanisms Underlying Obesity-Associated Pulmonary Dysfunction: Role of Adipose Tissue Mediators

Thanks to the discovery of adipokines, adipose tissue has been identified as a central node in the interorgan-crosstalk network and mediates the regulation of multiple organs and tissue types, including those of the respiratory system [1]. As mediators between adipose tissue and target organs, adipokines are a key factor in obesity-induced metabolic disorders [11]. Obesity-related inflammation and insulin resistance are exacerbated by adipokine overproduction in VAT, which has an autocrine and paracrine role in crosstalk with endothelial cells, fibroblasts, and immune cells [68]. In morbid obesity, the release of pro-inflammatory adipokines from dysfunctional adipose tissue causes low-grade systemic inflammation that is responsible for an increased risk of developing insulin resistance, hepatic steatosis, cardiometabolic disease, and pulmonary dysfunction, as previously discussed [28].

### 5.1. Leptin, Adiponectin, and Pulmonary Dysfunction

Leptin and adiponectin are two of the most abundant and well-studied adipokines, with a known role in lung function regulation [16,32,69,70,71,72,73,74].

Leptin is a hormone secreted by adipose cells, and its receptor is expressed both centrally in the hypothalamus and peripherally in several tissue [75,76]. Leptin can also be expressed in the lung by adipose-like lipofibroblasts and has a known role in regulating pulmonary surfactant production [77]. The central action of leptin promotes satiety and body weight reduction, while its action on adipose tissue increases glucose utilisation and lipolysis [75,76]. Paradoxically, hyperleptinaemia is common in obese subjects, and circulating levels of leptin are proportional to adipose tissue mass [78]. However, due to a feedback mechanism regulated by cytokine signalling 3 (SOCS3) and protein tyrosine phosphatase 1B (PTP1B), persistent leptin stimulation in obesity can lead to impaired leptin signalling and subsequent inability to mediate its anorexigenic effects, a state referred as ‘leptin resistance’ [76]. Therefore, we hypothesised that enhanced circulating leptin concentrations, particularly in subjects with obesity or type 2 diabetes mellitus, may contribute to several comorbidities, such as renal injury or cardiovascular disease [79,80]. Despite the precise mechanism not yet being known, leptin appears to be involved in the development of glomerulosclerosis through a paracrine TGF-β pathway (between glomerular endothelial and mesangial cells), which promotes the deposition of extracellular matrix and proteinuria [80]. Similarly, leptin appears to promote cardiac fibrosis and vascular dysfunction by increasing expression levels of profibrotic factors, such as collagen I, TGF-β, and connective tissue growth factor [80]. Another organ whose function is affected by leptin is the lung, where the leptin receptor is highly expressed. As a matter of fact, it has been shown that there is a significant association between genetic variants of the leptin receptor and lung function decline in patients with COPD [73]. Furthermore, the relationship between leptin and lung disease pathogenesis involves excess adipose tissue. It has been shown that serum leptin levels are strongly correlated with both body fat and lung function, and that high leptin levels promote OHS [81].

Adiponectin is another adipose-derived secreted hormone, with a critical role in regulation of whole-body energy homeostasis [82,83,84]. This adipokine is recognised by two different receptors, adipoR1 and adipoR2, which are expressed by several tissue types, including muscle, adipose, and lung [83,84]. Both receptors exert similar beneficial metabolic effects due to their ability to promote systemic glucose homeostasis, insulin sensitivity in adipose tissue, fatty acid oxidation in muscle, and anti-inflammatory effects in the lungs [85,86]. Contrary to leptin, adiponectin levels appear to be decreased in subjects with impaired lung function and obesity [82].

Both leptin and adiponectin have a direct role in regulating adipose tissue function [75,86], and serum levels of these adipokines are directly and inversely correlated with BMI in humans [87]. The ratio between adiponectin and leptin serum levels has been suggested as a novel biomarker index of dysfunctional adipose tissue. This ratio has been found to be significantly reduced in patients with insulin resistance, metabolic syndrome, and high levels of C-reactive protein (CRP), a protein found to be increased during acute inflammation [88]. Moreover, correlations between these metabolic dysfunctions and the adiponectin/leptin serum ratio appear to be stronger than those observed with adiponectin or leptin serum levels alone [88], likely because both adipokines are involved in the low-grade systemic inflammation present in obesity. As we discussed previously, systemic inflammation in obesity could affect other organs besides adipose tissue (i.e., increasing the risk of cardiovascular disease). Similarly, there is evidence that dysfunctional adipose tissue and systemic inflammation induced by obesity can contribute to airway inflammation and pulmonary disease [17]. In this context, several studies have shown that pulmonary dysfunction and metabolic disorders in patients with COVID-19 were directly correlated with excess VAT mass [89,90]; in contrast, an inverse correlation was observed with the adiponectin/leptin serum ratio [91]. These findings suggest that an altered adipokine-secretion profile from dysfunctional VAT in obese subjects could promote lung function decline in patients with COVID-19. Similar correlations have been observed in asthmatic patients with obesity as compared with obese subjects without asthma, where a lower adiponectin/leptin mRNA expression ratio was observed in adipose tissue in addition to increased serum markers of inflammation and altered alveolar macrophage function [17]. Bariatric surgery in asthmatic obese subjects reversed inflammation and recovered macrophage function, suggesting a relationship between excess adipose tissue, pulmonary disease pathogenesis, and lung immune function [17]. It was not surprising that a lower adiponectin/leptin serum ratio was also directly associated with obesity and airway inflammation exacerbation in patients with COPD [72,92,93].

Considering this evidence, recent in vivo and in vitro studies have explored the relationship between levels of these adipokines and lung diseases, revealing that leptin and adiponectin can directly regulate lung inflammation and immune function [69,94,95,96,97]. As a matter of fact, it has been observed that adipoR1 is expressed in immune regulatory T cells (Tregs) in the lungs and modulates the immune response during pulmonary inflammation [69]. Tregs are capable of suppressing the immune response, and their development is reciprocally interconnected with that of T-helper 17 (Th17) cells, which instead amplify the immune response and inflammation [94]. Therefore, the balance between these two cell populations is essential for the regulation of the immune system. In vivo studies on mice showed that activation of adipoR1 in lung Tregs promoted their proliferation and activity, thereby suppressing the immune response and inflammation [95]. However, this mechanism was impaired in obesity due to downregulation of adipoR1 expression, reducing Treg activity and promoting airway inflammation [69]. It appears that leptin exerts the opposite effect, inhibiting proliferation and activity of Tregs and promoting that of Th17 cells in patients with COPD and negatively impacting lung disease progression [96]. Moreover, administration of leptin to Th2 immune cells from the lungs of mice promoted production of pro-inflammatory cytokines IL-4, IL-5, and IL-13, while leptin deficiency in mice attenuated the inflammation induced by allergens [97]. These findings suggest that adipose tissue dysfunction induced by obesity could alter the adipokine profile (i.e., the adiponectin/leptin ratio), leading to an imbalance in the Treg/Th17 ratio, increasing the Th2 response, and increasing secretion of pro-inflammatory cytokines into the lung, leading to an increased risk of developing respiratory diseases.

### 5.2. Impairment of the NLRP3 Inflammasome in Obesity Leads to Pulmonary Dysfunction

Emerging evidence has indicated that the NLRP3 inflammasome, a critical component of the innate immune system, could play a key role in promoting lung disease in obesity [98,99,100,101]. This multimeric protein is a master regulator of inflammation and is found to be activated in both dysfunctional adipose tissue and the lungs of obese subjects [21,22,24]. Its main function is to trigger the cleavage of the cytokine precursors pro-interleukin-1β (pro-IL-1β) and pro-IL-18 in their active, pro-inflammatory forms, IL-1β and IL-18, respectively. This cleavage is mediated by caspase 1, a converting enzyme that is recruited following NLRP3 activation [99].

The NLRP3 inflammasome can be activated by both pathogen-associated molecular patterns (PAMPs) and danger-associated molecular patterns (DAMPs) [99]. Among the DAMPs, saturated fatty acids and other dietary metabolites derived from chronic nutrient excess have a key role in regulating NLRP3 activation in obesity, promoting insulin resistance and metabolic alterations [102,103]. It has been showed that NLRP3 expression and activity are involved in adipose tissue dysfunction and inflammation in obesity, in addition to their involvement in Treg/Th17 imbalance [104] and with lung health in patients with COPD, asthma, rheumatoid lung disease, and COVID-19 [22,105,106,107]. In particular, in vivo studies have shown that adipose tissue dysfunction can promote airway hyperreactivity, a cardinal feature of asthma that manifested depending on NLRP3 activity [101,108]. Development of asthma was shown to be induced by obesity, as mice fed a high-fat diet (HFD) rapidly developed insulin resistance, hepatic steatosis, airway hyperreactivity, and increased expression of NLRP3 and IL-1β in adipose tissue and the lung as compared with lean mice fed chow [101,108]. Moreover, mRNA expression levels of IL-1β and NLRP3 in the VAT of mice correlated with body weight and adiposity [108]. However, when NLRP3 was genetically depleted, mice fed an HFD showed a significant reduction in airway hyperreactivity, insulin resistance, and hepatic steatosis as compared with their wild-type littermates [101,108], indicating that the inflammasome complex has a key role in the development of these obesity-associated complications.

In addition to PAMPs and DAMPs, other regulators of NLRP3 activity have been discovered. Among these, leptin and adiponectin appear to play a role in modulating NLRP3-induced inflammation in the lung [109,110].

It is known that adiponectin can limit lung injury originating from endothelial cell damage, and more recently, Shah et al. have shown that a lack of adiponectin induces mitochondrial dysfunction and activates the NLRP3 inflammasome in lung endothelium [109]. It has been observed that dysfunctional mitochondria influence airway contractility and are involved in the pathogenesis of COPD in obesity [111]. Moreover, the findings that activated NLRP3 colocalised with damaged mitochondria in cells undergoing mitophagy blockade [112] and that oxidised mitochondrial DNA was a direct ligand and activator of NLRP3 [113] led to the suggestion that the inflammasome could be the missing link between mitochondrial dysfunction and lung disease. Adiponectin deficiency in mice has been reported to decrease mitochondrial biogenesis and impair mitochondrial function in the lung [109].

Mitochondrial function is also dependent on the turnover of damaged mitochondria through a process termed ‘mitophagy’ [55]. Of note, genetic depletion of adiponectin in mice decreased mitophagy, increased the accumulation of mitochondrial reactive oxygen species (mtROS), and enhanced NLRP3 inflammasome activation in lung endothelium as compared with wild-type mice. These findings were accompanied by evidence of pulmonary inflammation, immune-cell infiltration, and injury of the lung tissue. Treatment with pyrroloquinoline quinone (PQQ), a small polyphenolic compound with known antioxidant properties on mitochondria, decreased both NLRP3 activation and the deleterious effects of adiponectin deficiency on lung tissue [109]. These findings suggest that adiponectin could promote lung function by reducing mitochondrial oxidative stress and subsequent inflammasome activation.

On the contrary, leptin appears to promote disruption of the mitochondrial membrane, production of mtROS and NLRP3, and caspase 1 mRNA expression when administered to a line of human bronchial epithelial cells, leading to increased secretion of the pro-inflammatory cytokine IL-1β by these cells [110]. Of note, treatment of these cells with the mitochondria-targeted antioxidant mitoquinone (mitoQ) significantly decreased the IL-1β and NLRP3 expression that was induced by leptin, suggesting that this adipokine could lead to lung damage through mtROS–NLRP3 signalling [110]. These in vitro findings are consistent with studies conducted in individuals with airway inflammation, where serum levels of both leptin and IL-1β were shown to increase with BMI and were thus higher in subjects with obesity [114].

Another mechanism of regulation of NLRP3 activity by leptin was observed in a macrophage-like cell line derived from mice, where leptin treatment increased NLRP3 expression and induced activation of caspase 1 in a dose-dependent manner, increasing cleavage of pro-IL-18 in its active form [29]. These effects were reduced when cells were treated with a caspase 1 inhibitor or when NRLP3 gene expression was silenced, demonstrating that leptin increased expression of these pro-inflammatory cytokines through an NLRP3–caspase 1 pathway [29]. Considering that activation of NLRP3 and secretion of IL-1β and IL-18 in alveolar macrophages contribute to pulmonary inflammation and injury [23], and that there is a known relationship between leptin, IL-18 serum levels, lung function, and body fat [30], further studies should be conducted to investigate the role of this adipokine in promoting maturation of NLRP3-derived pro-inflammatory cytokines in the context of obesity-associated pulmonary diseases.

Therefore, it appears that lung diseases are promoted by the secretome from dysfunctional adipose tissue, the altered adipokine profile of which drives NLRP3 inflammasome activation and subsequent production of IL-1β and IL-18. Accordingly, levels of pro-inflammatory cytokines, including IL-1β and IL-18, have been found to be increased in subjects with obesity or respiratory diseases [115,116,117]. In contrast, we showed that correction of IH in patients with OSAS reduced levels of pro-inflammatory cytokines in the serum and tended to reduce NLRP3 mRNA expression in adipose tissue, suggesting that the relationship between lung function and systemic inflammation could be bidirectional and modulated by the inflammasome [5]. In contrast, we showed that correction of IH in patients with OSAS reduced levels of pro-inflammatory cytokines in the serum and tended to reduce NLRP3 mRNA expression in adipose tissue, suggesting that the relationship between lung function and systemic inflammation could be bidirectional and modulated by the inflammasome [5]. This hypothesis was supported by other studies that showed that induction of IH promoted adipose tissue inflammation in mice [118], increased leptin expression in human adipocytes [119], and activated NLRP3–IL-1β signalling in M1 pro-inflammatory macrophages [120]. In accordance, it has been shown that obese subjects with OSAS have increased levels of leptin and higher NLRP3 activity [121,122]. Conversely, NLRP3 deficiency protected against IH in a mouse model of OSAS [123]. Taken together, these findings suggest a vicious circle where obesity favours OSAS and IH promotes adipose tissue inflammation.

Focusing research on subjects with both airway inflammation and obesity, recent studies have shown that IL-1β serum and sputum levels are correlated with BMI and are an important risk factor for the exacerbation of symptoms [114,117]. However, a broad spectrum of other cytokines has also been found to be altered in obesity-related lung diseases. Among these, IL-5, IL-17, IL-13, IL-23, and IL-6 are pro-inflammatory cytokines upregulated in patients with pulmonary diseases [124,125,126]. Of note, levels of these cytokines also increase with increasing BMI in subjects with both obesity and airway inflammation, suggesting that obesity could be an important risk factor for the exacerbation of lung symptoms. IL-5 and IL-17 expression in the sputum and IL-13 and IL-17 expression in the serum appear to be upregulated in obese asthmatic patients as compared with lean asthmatic patients [127,128,129,130]. In patients with OSAS, there was strong positive correlations between IL-6, IL-17, IL-23, and leptin serum levels and BMI, body fat, and markers of metabolic dysfunction [130]. Moreover, IL-6 serum levels and IL-26 sputum levels were higher in obese patients as compared to lean patients with COPD [131,132]. In addition to pro-inflammatory cytokines, altered levels of cytokines with anti-inflammatory properties have been found in obese patients with lung diseases. Among these, IL-10 is an anti-inflammatory cytokine produced by Tregs in the lung [95]; its serum levels were found to be reduced in patients with OSAS and obesity [133].

Several studies have shown that expression of these cytokines (IL-17, IL-23, IL-13, IL-5, IL-6, IL-26, and IL10) are modulated by NLRP3–IL-1β signalling in the lung [5,134,135,136,137]. Exogenous administration of IL-1β in mice increased secretion of IL-23 from bronchial epithelial cells, the paracrine action of which exacerbated the immune response and IL-17 expression from Th17 cells [135]. IL-23 and IL-17 production was downstream of IL-1β signalling, as expression of these cytokines was abrogated following genetic depletion of the IL-1β receptor (IL1β-R); in contrast, genetic depletion of IL-17 did not affect expression of IL-1β [135]. Further studies in mice showed that the NLRP3 inflammasome was required for asthma development and expression of the cytokines IL-13, IL-5, and IL-6 in obese mice fed an HFD [134]. Moreover, depletion of the NLRP3 gene prevented airway inflammation and expression of these pro-inflammatory cytokines [134]. In vitro studies in human immune cells showed that IL-1β was responsible for IL-26 induction by Th17 cells [136], which in turn appeared to promote IL-6 secretion in human monocytes [138]. This inflammatory response required NLRP3 activation and function, as it did not occur when monocytes were pretreated with an inflammasome inhibitor [138]. In vivo studies in the lungs of mice have shown that this inflammatory response is negatively regulated by IL-10, which exerts anti-inflammatory effects on NLRP3 inflammasome signalling through a negative feedback mechanism. [137]. Taken together, these findings lead to the belief that the inflammasome could be the main inducer of a pro-inflammatory cytokine cascade that is responsible for immune response imbalance and the onset of pulmonary diseases in obesity. This hypothesis is in accordance with evidence that leptin and adiponectin have a role in regulating Th17 and Th2 immune responses in lung diseases [95,97,139,140], in addition to the expression of pro-inflammatory cytokines and NLRP3 activity, as previously discussed [109,110]. This suggests a common pathway that links the dysfunctional adipose tissue to lung inflammation and injury (Figure 1).

## 6. Therapeutic Targets to Prevent Pulmonary Dysfunction in Obesity

Considering the relationship between obesity and lung health, weight loss represents a valid option to improve pulmonary function. It has been demonstrated that patients with interstitial lung disease and severe obesity who underwent bariatric surgery had a dramatic improvement in pulmonary function a year after bariatric surgery [141]. Even with lower impact on weight reduction (at least 2 kg body weight reduction) than bariatric surgery, nutrition-education and dietary restrictions induced a significant improvement in main outcomes of pulmonary performance (forced vital capacity—FVC, diffusing capacity of the lung for carbon monoxide—DLCO, and functional residual capacity—FRC), after short-term intervention in obese patients with interstitial lung disease [142]. When low caloric intake was combined with weight loss medication (sibutramine and orlistat), with a reduction in BMI of 5.3 kg/m^2^, a significant improvement in FVC was observed in the interventional group compared to the control one [143] (Figure 2). However, when considering interventional studies for weight reduction through nutritional counselling and/or low caloric intake and/or weight loss medication [144], immunological parameters and pro-inflammatory markers were not changed in patients with asthma-related pulmonary diseases, except for CRP [145].

Another therapeutic approach could be to reduce systemic inflammation in morbid obesity (Figure 2). In this regard, the crosstalk between dysfunctional adipose tissue and the lungs involves inflammatory mediators, such as the adipokines leptin and adiponectin, the inflammasome NLRP3, the inflammasome-derived cytokines IL-1β and IL-18, and their downstream cytokine effectors IL-23, IL-17, IL-26, IL-6, IL-13, and IL-5 [5,128,129,130,131]. These inflammatory endpoints could be used as therapeutic targets for the treatment of pulmonary disease in obese subjects.

Considering the favourable metabolic properties of adiponectin, a synthetic ligand able to activate adipoR1 and adipoR2, named AdipoRon, was developed [146]. Adiponectin receptors are extensively expressed in cell lines of distinct origins; therefore, a pharmacological agonist would be able to act in a pleiotropic manner on different target tissue types [84]. When evaluated for its antidiabetic properties in mice, AdipoRon promoted fatty acid combustion in the liver, increased insulin sensitivity in muscle, decreased expression of pro-inflammatory cytokines, and ameliorated adipose tissue dysfunction [146,147]. Moreover, it is known that adiponectin receptors are expressed in lung epithelial cells, Tregs, and macrophages, regulating airway inflammation and the immune response [31,32,83]. Accordingly, administration of adiponectin or AdipoRon into human Tregs significantly increased their expression of the anti-inflammatory cytokine IL-10 [95]; in macrophages from human lung explants, expression of pro-inflammatory cytokines was reduced in a concentration-dependent manner [32]. To improve solubility and bioavailability, an AdipoRon analogue containing amphiphilic polyethylene glycol (PEG) chains has been developed and has shown improved anti-lipotoxic properties [148]. To date, adiponectin agonists have not yet been tested in humans; however, in vivo and in vitro evaluation of their use in the treatment of pulmonary dysfunction induced by obesity is promising.

As previously discussed, the ability of adiponectin and leptin to modulate inflammasome activity may depend on a mechanism mediated by mtROS. MitoQ, a mitochondria-target antioxidant, decreased inflammation in an in vitro model of a pulmonary vascular endothelial barrier [149], and these results were successfully translated in vivo, where this drug reduced lung inflammation in mice with AHR [150]. Given these promising outcomes, MitoQ is currently being evaluated for the treatment of metabolic dysfunction in asthma [151].

Another potential therapeutic target is the NLRP3 inflammasome. Tranilast, a drug currently approved in Japan for the treatment of asthma [152], has been recently rediscovered for its ability to inhibit the NLRP3 inflammasome [153], paving the way for its use in the treatment of other NLRP3-driven diseases. In vitro studies on human macrophages have shown that treatment with tranilast reduced IL-1β secretion via direct interaction with the NLRP3 protein, inhibiting its oligomerisation and activation [142]. Of note, tranilast induced weight loss [154] and exhibited antidiabetic properties in mice via NLRP3 inhibition [155], suggesting that this drug might reduce obesity-associated comorbidities. In light of this evidence, tranilast is under evaluation as a treatment for patients with COVID-19 [156] whose lung damage appears to be promoted by obesity through an NLRP3-driven mechanism [100,157].

Considering the rediscovered therapeutic potential of tranilast via inflammasome modulation, several researchers have attempted to develop a more selective inhibitor. Among these, MCC950 is a diarylsulfonylurea-containing compound that is considered one of the most potent and selective inhibitors of the inflammasome; it is able to block IL-1β production by abrogating NLRP3 oligomerisation [158]. When tested in mice, treatment with MCC950 attenuated the increase in body weight induced by an HFD, reduced IL-1β concentrations in adipose tissue, prevented insulin resistance, and reduced airway inflammation induced by obesity [159,160]. Unfortunately, in a human phase II clinical trial, MCC950 caused liver toxicity and was discontinued [161]. More encouraging results came from OLT1177 (dapansutrile), a β-sulfonyl nitrile and the first NLRP3 inhibitor shown to be safe in humans [162]. Like tranilast and MCC950, this molecule binds directly to the NLRP3 protein to inhibit its oligomerisation and activity [162]. In lipopolysaccharide (LPS)-stimulated human macrophages, OLT1177 reduced the release of pro-inflammatory cytokines IL-1β and IL-18; in mice with allergic asthma, OLT1177 ameliorated symptoms via NLRP3 inhibition [162,163]. These promising results encouraged implementation of OLT1177 in a human clinical trial for the treatment of patients with COVID-19, which is ongoing [164].

A key regulator of IL-1β and IL-18 maturation is caspase 1; therefore, inhibiting its activity could represent a promising treatment approach. In vivo studies on mice with lung infections showed a significant reduction in plasma levels of pro-inflammatory cytokines IL-1β and IL-18 following administration of GKT137831 (setanaxib), an inhibitor of the NADPH oxidase 4 (NOX4) enzyme involved in caspase 1 activation [165]. NOX4-mediated inhibition of caspase 1 activation reduced adipocyte dysfunction in vitro, ameliorated propensity for obesity in HFD-fed mice, and attenuated pulmonary hypertension in mice exposed to hypoxia, thereby preventing lung dysfunction [165,166]. Given these promising results, GKT137831 is currently being tested in patients with idiopathic pulmonary fibrosis and patients with type 2 diabetes [167,168].

Considering that IL-1β, IL-17, IL-13, and IL-5 are involved in the cytokine cascade downstream of NLRP3, monoclonal antibodies for these cytokines have been developed to block their pro-inflammatory effects. Although this approach produced a positive response in in vivo studies for the treatment of asthma [98,169,170], conflicting results were obtained when these therapies were translated from animals to humans. In a clinical trial presented as a conference abstract, treatment with canakinumab, a human anti-IL-1β monoclonal antibody, attenuated response to allergens in patients with mild asthma [171]. Despite these encouraging preliminary data, no further studies on asthmatic patients were conducted, and a clinical trial in patients with COPD was inconclusive [172]. Another approach to targeting IL-1β signalling is to block its receptor instead of its ligand. Anakinra (MEDI8968), an IL-1β-R agonist, consistently reduced blood neutrophil, CRP, and fibrinogen levels in COPD patients; however, it failed to improve lung function [173]. However, in patients with COVID-19 pneumonia, both canakinumab and anakinra significantly reduced inflammation and improved lung function without any severe adverse effects [172,174]. Mixed results were also obtained with benralizumab, an anti-IL-5 receptor antibody that ameliorated symptoms in patients with asthma [175], but was ineffective in patients with COPD [176]. Moreover, antibodies against IL-13 and IL-17, named tralokinumab (CAT-354) and CNTO 6785, respectively, did not improve asthma outcomes or reduce COPD exacerbations when used alone. Therefore, a bispecific antibody targeting both IL-13 and IL-17 simultaneously, named BITS7201A, was developed to assess the combined blockage of these two cytokines [177]. Unfortunately, evaluation in humans was terminated early, due to a high incidence of anti-drug antibodies and risk of immunogenicity, impeding the acquisition of sufficient data on its therapeutic efficacy [177].

The partial failure of anti-cytokine therapies to significantly improve clinical outcomes in patients with lung diseases could be explained by the functional redundancy of inflammatory signalling pathways. Although canakinumab can effectively bind to IL-1β [178], it should be considered that the inflammasome also affects production of other cytokines, such as IL-18 [99,179], whose levels have been found to be strongly associated with lung function in patients with COPD and asthma [180,181] and with inflammation and insulin resistance in obese individuals [115]. Moreover, even in those inflammatory diseases where canakinumab has been shown to be effective (e.g., arthritis), a residual inflammatory risk associated with IL-18 and IL-6 levels has been observed [179]. It could be hypothesised that this residual inflammatory risk is more pronounced in lung diseases due to the involvement of multiple inflammatory signalling pathways [182]. Moreover, the contribution of IL-13, IL-17, and IL-5 alone to airway inflammation could be only partial; therefore, targeting them individually may be therapeutically insufficient in preventing lung dysfunction. With these considerations in mind, a promising approach could be to develop combined therapies or bispecific antibodies for these cytokines [177]. Another therapeutic target could be the NLRP3 inflammasome, whose activity is upstream of a pro-inflammatory cytokine cascade with harmful effects that affects both adipose tissue and lungs in obesity [17]. Therefore, blocking the activation of the inflammasome would prevent obesity-associated comorbidities, ameliorating lung inflammation and function in addition to metabolic health (Figure 2).

## 7. Conclusions

In addition to mechanical ventilatory constraints, excess adipose tissue in obesity can lead to ectopic deposition of fat in visceral depots, promoting metabolic disorders and altering the adipokine-secretion profile. In particular, the adipokines leptin and adiponectin appear to have a role in the onset of obesity-derived pulmonary diseases by affecting systemic inflammation, immune Treg activity, and the Th17 and Th2 responses. It has been shown that these adipokines can also modulate the activity of the NLRP3 inflammasome, a multimeric complex with a key role in inflammatory processes that is found to be activated in both the adipose tissue and lungs of obese subjects. Moreover, NLRP3-derived cytokines (e.g., IL-1β, IL-18) appear to be the main inducers of a cytokine cascade (involving IL-23, IL-17, IL-26, IL-6, IL-13, and IL-5) that is responsible for the development of pulmonary diseases in morbidly obese subjects (Figure 1). Therefore, pharmacological strategies to target these pro-inflammatory mediators has attracted substantial interest from researchers. However, recent clinical trials on monoclonal antibodies, such as canakinumab and anakinra, have returned mixed results in terms of clinical benefit, with significantly reduced lung inflammation in patients with COVID-19 and limited improvement in patients with COPD and asthma. This partial failure could be explained by the functional redundancy of inflammatory signalling pathways activated by several cytokines in obesity and lung diseases; most of the antibodies currently tested in humans were developed to target a single cytokine. A more effective approach could be to develop combined therapies or bispecific antibodies to target multiple cytokines. Another therapeutic target could be the NLRP3 inflammasome, whose activity is upstream of both IL-1β and IL-18. Inhibitors of NLPR3 activity would prevent systemic inflammation induced by obesity and related comorbidities, including pulmonary dysfunction.

These findings indicate that obesity and lung dysfunction are closely related. The search for the optimal therapeutic solution should involve multiple inflammatory pathways, focusing on the pathways that are shared between dysfunctional adipose and lung tissue to prevent both metabolic disorders and pulmonary diseases. Since adipose tissue dysfunction negatively affects lung health, obesity could be an important, suitable target for respiratory disease prevention. Therefore, obesity treatments that are able to reduce adipose tissue and induce weight loss, such as lifestyle interventions, pharmacotherapy, and bariatric surgery, could represent another approach to treating pulmonary diseases.

## Figures and Tables

**Figure 1 ijms-23-07349-f001:**
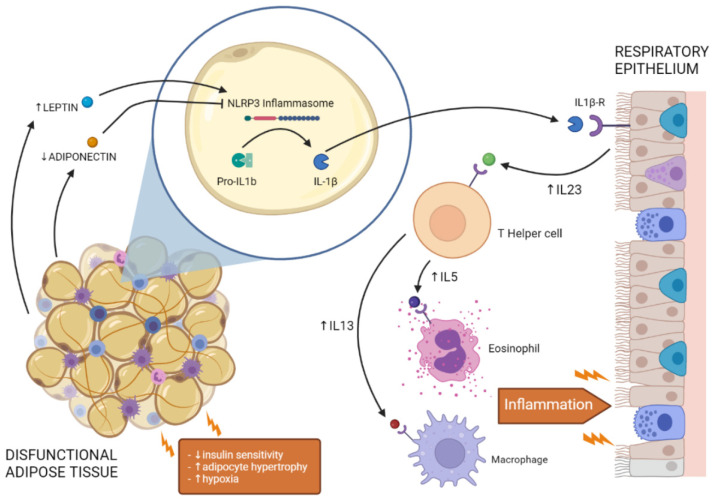
Crosstalk between dysfunctional adipose tissue and respiratory epithelium in the lung. Insulin resistance, adipocyte hypertrophy, and hypoxia alter adipokine secretion from dysfunctional adipose tissue, promoting NLRP3 inflammasome activation and IL-1β secretion. Activation of IL-1β signalling in the lung increases the expression of pro-inflammatory cytokines (e.g., IL-23, IL-5, IL-13), promoting the activity of immune cells (e.g., T-helper cells, eosinophils, macrophages) and leading to lung inflammation.

**Figure 2 ijms-23-07349-f002:**
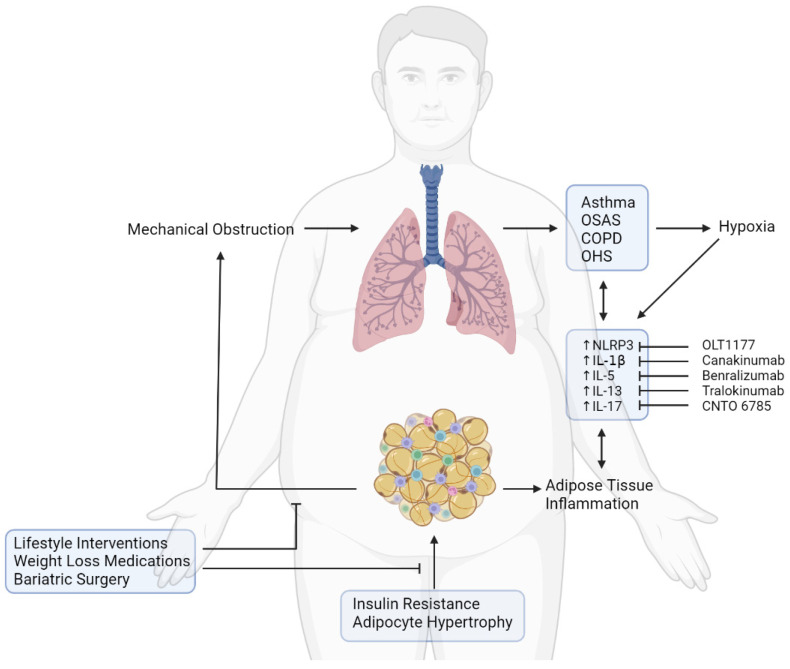
Effects of obesity on pulmonary function and therapeutic strategies for the treatment of lung diseases. Insulin resistance and adipocyte hypertrophy induce adipose tissue inflammation. Excess of adipose tissue is responsible for mechanical obstruction of lung airways, promoting development of asthma, OSAS, COPD, and OHS, which are responsible for hypoxia. Both hypoxia and adipose tissue inflammation foster activation of the NLRP3 inflammasome, increasing levels of pro-inflammatory cytokines (IL-1β, IL-5, IL-13, IL-17) and promoting development of lung diseases. Improvement in the respiratory function in people with obesity and lung disease could be achieved through therapeutic intervention for weight loss or through pharmacological strategies able to target the inflammasome and its downstream cytokines.

## Data Availability

Not applicable.

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
