# Peer review of "Adipose Tissue Inflammation and Pulmonary Dysfunction in Obesity"

_ijms, 2022, doi:10.3390/ijms23137349_

Round 1

Reviewer 1 Report

In this interesting rewiev, Giuseppe Palma and coll. exstensively described obesity as risk factor for altering the functionality of various organs, including the lungs, throught mechanical ventilatory defects, as well as metabolic disorders due to adipose tissue dysfunction; it's now acclarated that adipose tissue is a real endocrine organ, promoting chronic stystemic inflammation.

have the authors evaluated the possible role of some specific mRNAs, despite the fact that in the literature there is a too extensive mention of possible mRNAs?

Reviewer 2 Report

In the review article entitled “Adipose Tissue Inflammation and Pulmonary Dysfunction in Obesity” authors have discussed about the effect of obesity on lungs and its function. They have summarized how the excess of adipose tissue causes various pulmonary associated diseases and the therapeutic targets for the treatment of pulmonary dysfunction caused by obesity. The review article is of great interest to readers and is well written. Few points to consider making this article stronger are:

1.     Authors could reframe or elaborate the sentence under section 2 “Obese people breathe at low lung volumes associated with limited expiratory flow because elastic recoil determines the calibre of the airways”

2.     In line 87 authors could replace the word “adipose tissue” with “obesity” or “excess of adipose tissue” .

3.     In line 132 the authors could replace “and that this process” by “which”.

4.     Authors could reframe the sentence “Systemic inflammation induced by VAT is due to its modality of expansion, which appears to occur differently than SAT; increased VAT mass in morbid obesity largely results from adipocyte hypertrophy due to the dimensional increase of intracellular lipid droplets, whereas hyperplasia is predominantly seen in SAT” (Line 140-142)

5.     Under section 3 “Adipose Tissue Dysfunction and Pulmonary Diseases” authors have mentioned proinflammatory cytokines but it would be of more interest if authors can specifically mention examples of proinflammatory cytokines that VAT secrete specially under obese conditions that lead to pulmonary diseases.

6.     The sentence “Recently, it was recognized that the gut microbiota has a role in the development of OSAS and obesity and might thus be a factor in the coexistence of OSAS and obesity” under section 4.2 , is kind of breaking a flow. The authors are discussing about comorbidities related to OSAS and obesity and suddenly this sentence comes in between.

7.     In section 4.2. “Macrophages in fat are likely the target cells for the effects of persistent IH, resulting in elevated inflammatory biomarkers, which suggests that OSAS and obesity may have synergistic effects” where authors are discussing the role of macrophages in causing OSAS under obesity condition, it would be great if authors can specifically indicate if M1 or M2 macrophage causes IH, as  obesity could lead to differential activation of M1 or M2 macrophage which secrete anti-inflammatory or pro-inflammatory cytokines.

8.     It is not clear what authors want to communicate in the sentence “Importantly, the management of COPD comorbidities, such as obesity, is associated with a significant portion of the disease burden and health-care utilization”, under section 4.3

9.     In section 5, line 315, authors can change or exclude word “Thanks to”

10.  In section 5, authors should add reference after first and second sentence.

11.  In section 5.1, authors should discuss how leptin excess lead to renal injury and cardiovascular diseases, could include the mechanism, how leptin causes or responsible for renal injury and cardiovascular diseases.

12.  Authors should reframe sentence “In this regard, it has been demonstrated that the leptin receptor is also expressed in the lung, and a significant association has been shown between genetic variants of this receptor and lung function decline in patients with COPD”, line 342 and 343.

13.  In line 346, authors can replace word “was” with “has been”.

14.  In line 347, authors should briefly describe briefly about adipokine “adiponectin” before suddenly bring it into context and if possible, start a new paragraph for discussing about the role of adiponectin.

15.  Authors could discuss if leptin and NLPR3 also regulate OSAS under obesity condition?

16.  In figure 1, the spelling of “hypoxia” is incorrect

17.  Authors could include the importance of weight loss or weight management as an important part for the treatment of the pulmonary diseases caused due to obese condition mentioned in this review article.

18.  In line 659, authors could replace the word “promoting” by “affecting or modifying”.

19.  The authors could reframe the sentence (line 679) “These findings suggest that obesity and pulmonary dysfunction should be considered related entities”.

20.  Authors could reframe the sentence (line 683) “The clear link between obesity and pulmonary disease suggests that obesity is an important, suitable target for respiratory disease prevention”.

21.  If authors could include the crux of their review article in graphical form would be of great interest and help readers to understand.
